# H3K27 modifications define segmental regulatory domains in the *Drosophila* bithorax complex

Sarah K Bowman[1,2†], Aimee M Deaton[1,2†], Heber Domingues[3], Peggy I Wang[1,2], Ruslan I Sadreyev[1,4], Robert E Kingston[1,2]*, Welcome Bender[3]*

[1]Department of Molecular Biology, Massachusetts General Hospital, Boston, United States; [2]Department of Genetics, Harvard Medical School, Boston, United States; [3]Department of Biological Chemistry and Molecular Pharmacology, Harvard Medical School, Boston, United States; [4]Department of Pathology, Harvard Medical School, Boston, United States

**Abstract** The bithorax complex (BX-C) in *Drosophila melanogaster* is a cluster of homeotic genes that determine body segment identity. Expression of these genes is governed by cis-regulatory domains, one for each parasegment. Stable repression of these domains depends on Polycomb Group (PcG) functions, which include trimethylation of lysine 27 of histone H3 (H3K27me3). To search for parasegment-specific signatures that reflect PcG function, chromatin from single parasegments was isolated and profiled. The H3K27me3 profiles across the BX-C in successive parasegments showed a 'stairstep' pattern that revealed sharp boundaries of the BX-C regulatory domains. Acetylated H3K27 was broadly enriched across active domains, in a pattern complementary to H3K27me3. The CCCTC-binding protein (CTCF) bound the borders between H3K27 modification domains; it was retained even in parasegments where adjacent domains lack H3K27me3. These findings provide a molecular definition of the homeotic domains, and implicate precisely positioned H3K27 modifications as a central determinant of segment identity.

*For correspondence: kingston@molbio.mgh.harvard.edu (REK); (+1) 617-432-1906 (WB)

†These authors contributed equally to this work

**Competing interests:** The authors declare that no competing interests exist.

**Reviewing editor**: Danny Reinberg, Howard Hughes Medical Institute, New York University School of Medicine, United States

## Introduction

Cells in higher organisms choose among developmental pathways in response to transitory cues. The choice must be remembered; this is often described as changing the epigenetic state. One prominent mechanism to fix such choices relies on the genes of the Polycomb Group (PcG). The members of this family have been well defined by genetic and biochemical studies (*Simon and Kingston, 2013*). The PcG proteins are present in all cells, but they impose long-term repression on different target genes in different lineages. PcG proteins form distinct complexes with distinct functions. These include the PRC1 family of complexes, which compact chromatin and ubiquitylate histone H2A, and the PRC2 complexes, which methylate histone H3 on lysine 27. The molecular details of how PcG functions are coordinated to lead to repression are poorly understood.

The prototypic targets of PcG action are the *Drosophila* homeotic genes that control segment identity. Indeed, a mutation in *Polycomb*, the founding member of the PcG, was discovered for its segmental transformations (*Lewis, 1947*). In *Drosophila melanogaster*, the homeotics lie in two clusters, the Antennapedia complex (Antp-C) and the bithorax complex (BX-C). The BX-C is the better-studied cluster, due largely to the effort of EB Lewis. Lewis showed that mutations that transform specific segments align on the genetic map in the order of the body segments that they affect (*Lewis, 1978*). The evolutionary conservation of this gene order, including that in the mammalian HOX complexes, suggests that chromosomal position dictates the expression pattern. Lewis used nested

**eLife digest** Like other insects, the body of the fruit fly is divided into three main parts—the head, the thorax and the abdomen—and each part, in turn, is made up of several smaller segments. The bithorax complex is a cluster of three genes that together control the identity of the segments that make up the back half of the fruit fly's body. This gene cluster has been studied for several decades and these studies have helped to further our understanding of how genetic information is accessed and used to make an animal's body plan.

Early on in a fruit fly embryo, stretches of DNA within the bithorax complex regulate where the complex's genes are switched on, and where they are switched off. Proteins called Polycomb group proteins then keep the silenced genes off, in part by adding small chemical marks to other proteins called histones. Most DNA in a cell is wrapped around histones, and the addition of such chemical marks causes the DNA to become more tightly packed. This prevents the bithorax complex genes from being accessed and switched on. It had previously been suggested that each segment might have a unique pattern of chemical marks on the bithorax complex histones, but evidence to support this idea was lacking.

Bowman et al. have now undertaken the technically challenging task of purifying the DNA and its histones from individual segments of fruit fly embryos. This revealed that segments closer to the embryo's head contain larger stretches of bithorax complex DNA covered with histones marked by the Polycomb group proteins. Bowman et al. also found that the coverage of chemical marks on the histones changed dramatically when one segment was compared to its neighboring segments. These sharp boundaries clearly outline which regulatory regions of the DNA are switched on and which are switch off; however the same pattern is not seen for the Polycomb group proteins themselves. Instead, within the bithorax complex, the pattern of these proteins is almost identical in different segments.

The challenge now is to understand how the chemical marks and the Polycomb group proteins work together to restrict access to DNA in such precise patterns. Also—since similar gene clusters control the development of the body plans of mammals—this, in turn, might help us to understand how the Polycomb group proteins perform similar functions in human development and disease.

deletions to show that successively more distal genetic functions of the BX-C are activated in successively more posterior segments along the body axis (*Lewis, 1981*). The genetic functions are now thought to encompass nine regulatory domains, each for a different parasegment. The regulatory domains control three homeobox-containing genes of the complex, *Ultrabithorax* (*Ubx*), *abdominal-A* (*abd-A*), and *Abdominal-B* (*Abd-B*) (*Maeda and Karch, 2009*). The regulatory domains have been roughly defined by mutant lesions (*Lewis, 1978*), and by enhancer traps in the BX-C, which show different anterior limits of expression (*Bender and Hudson, 2000*). The molecular features that define these regions, as well as the precise locations of their boundaries, have not been defined.

Both PRC1 and PRC2 are required to maintain repression of homeotic genes in appropriate parasegments, so these complexes may generate segment-specific chromatin features in the regulatory domains of the BX-C. We initially evaluated H3K27me3 patterns because they reflect the central function of PRC2. The entire BX-C is heavily marked with H3K27me3 in whole embryos (*Schuettengruber et al., 2009*; *Nègre et al., 2011*). This histone modification is necessary for maintaining repression of BX-C genes (*Pengelly et al., 2013*), yet these genes are not repressed throughout the body. It seemed likely that parasegment-specific patterns of histone modification might be obscured when all the parasegments are pooled for analysis. Indeed, *Papp and Müller (2006)* observed loss of the K27me3 mark across the *Ubx* transcription unit in haltere and third leg imaginal discs, tissues where *Ubx* is transcribed. These discs include cells of two parasegments (PS5&6), and so it was not clear how the K27me3-free regions might correlate with transcription units or with the genetic regulatory domains. The issue could be resolved if H3K27me3 patterns could be studied in single parasegments.

## Results

The chromatin features of the BX-C have not been studied in individual parasegments because of the technical challenges of cell isolation and molecular analysis on small samples. To address this,

we marked single parasegments with a combination of Gal4 and Gal80 drivers, and developed a nuclear sorting-chromatin immunoprecipitation-sequencing protocol. Gal4, a transcriptional activator, was expressed in a series of parasegments with a defined anterior boundary, and Gal80 was expressed in a pattern shifted one parasegment more posterior. Gal80 binds to and inactivates Gal4, leaving Gal4 activity in a single parasegment (*Figure 1A*). Gal4 and Gal80 expression domains were established either by enhancer trapping or by selecting an enhancer to achieve the desired expression pattern (*Figure1—figure supplement 1*). Various combinations of Gal4 and Gal80 drivers were combined genetically to limit Gal4 activity to each of parasegments 4, 5, 6, and 7, which approximately correspond to the second thoracic through the second abdominal segments (*Figure 1B*, *Figure1—figure supplements 1 and 2*; 'Materials and methods' for details).

For isolation of marked nuclei, we used Gal4 to drive an mCherry-RanGAP fusion protein, from the INTACT system (*Deal and Henikoff, 2011*; *Steiner et al., 2012*). Staged embryos with single parasegment stripes of mCherry were formaldehyde-fixed and disrupted, and their nuclei were sorted on a FACS instrument (*Figure1—figure supplement 1C*). DNA enriched by chromatin immunoprecipitation (ChIP) was prepared for paired-end sequencing, using a picogram-scale library protocol (*Figure 1—figure supplement 1D*; *Bowman et al., 2013*).

The largest differences in H3K27me3 enrichment between parasegments occurred at the BX-C (*Figure 2A*, *Figure 2—figure supplement 1*). While the H3K27me3 profiles in our four adjacent parasegments were very similar elsewhere in the genome (*Figure 2—figure supplement 1*), the BX-C exhibited a striking 'stairstep' pattern. Large expanses of H3K27me3 enrichment were lost as we moved from anterior to posterior along the body axis (*Figure 2C–F*), revealing the locations of the regulatory domains. This is reminiscent of the temporal activation pattern of the Hox-D cluster in mouse, which correlates with loss of H3K27me3 (*Soshnikova and Duboule, 2009*). Our analysis of single parasegments locates distinct domain borders where K27me3 status changes abruptly. The contrast between the whole embryo profile (*Figure 2B*) and the single segment profiles (*Figure 2C–F*) shows how isolated cell types can identify chromatin changes linked to cell identity.

In each isolated parasegment, the extent of K27me3 coverage was largely consistent with genetic studies indicating the location of BX-C active and repressed regulatory domains. The PS4 H3K27me3 pattern is mesa-like across the complex (*Figure 2C*), with only a few narrow gaps at nucleosome-free regions (*Mito et al., 2005*). The BX-C is almost completely repressed in PS4, with only a few neuronal cells on the midline showing UBX expression in embryos. The PS5 regulatory domain, associated with the *bithorax* (*bx*) mutant lesions, is defined by the low level of H3K27me3 modification across the leftmost (proximal) 92 kb of the complex (*Figure 2D*). There is a sharp transition to full H3K27me3 coverage at a position upstream of the *Ubx* transcription start site. The low residual level of H3K27me3 marks in the 92 kb PS5 domain is likely due to partial contamination from more anterior parasegments in our preparation of PS5 nuclei (*Figure 1—figure supplement 2*). In PS6, the leftmost 137 kb of the BX-C is clear of the K27me3 mark (*Figure 2E*); the additional K27me3-clear region reveals the PS6 domain. The PS6 regulatory domain is the one best defined by genetic analysis, with a series of rearrangement breakpoints giving *bithoraxoid* (*bxd*) phenotypes (*Bender and Lucas, 2013*). The transition point to full K27 methylation coincides with the genetically defined *Fub* border between the PS6 and the PS7 regulatory domains (*Bender and Lucas, 2013*). In PS7, an additional 36 kb was depleted of H3K27me3 (*Figure 2F*). This region includes the *abd-A* transcription unit, and is associated with the *infraabdominal-2* (*iab-2*) class of genetic mutations. We observed a reproducible mound of H3K27me3 in PS7 nuclei close to the *Ubx* promoter (arrow in *Figure 2F*). This may reflect repression of *Ubx* by *abd-A* in this parasegment (*Karch et al., 1990*), mediated by the PcG.

The regulatory domains defined by the sharp transitions in K27me3 enrichment do not correspond in every detail to expectations from genetic studies. Prior mapping of domains has relied upon random insertion of transgenic reporter elements into the BX-C to interrogate local enhancer activity. These enhancer traps show sharp anterior edges to their expression patterns, with the anterior-most labeled parasegment corresponding to the domain in which the insertion resides (*Bender and Hudson, 2000*). In the current study, the border between the PS5 and PS6 domains, defined by the discontinuity in K27me3 enrichment in PS5 nuclei, lies 14 kb upstream of the *Ubx* promoter. However, transgenic reporter elements inserted between the *Ubx* promoter and the K27me3-defined border (presumably part of the PS5 regulatory domain) show patterns with a PS6 anterior limit. Perhaps the promoters of these reporters interact with the *Ubx* promoter in distinctive ways. There was also prior evidence of an active BX-C domain in PS4. An enhancer trap 3 kb downstream of the *Ubx* start site

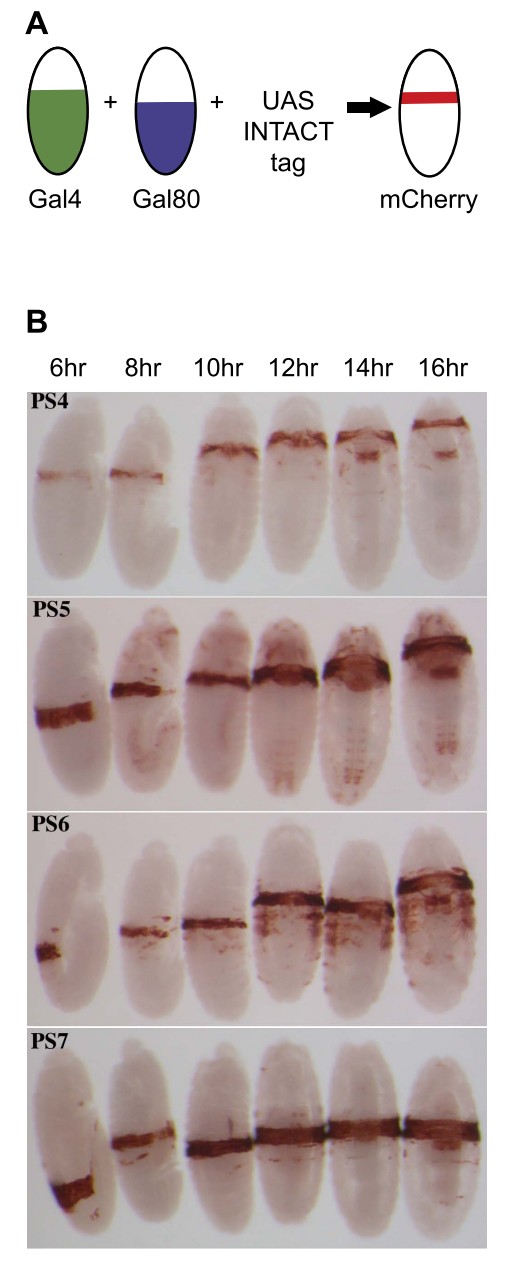

**Figure 1**. Marking single parasegments. (**A**) Drivers for the Gal4 activator and the Gal80 repressor, each with a different anterior limit, are combined genetically. Gal4 activity is thus limited to a single parasegment, and is used to activate transcription of a fluorescent nuclear envelope protein. (**B**) Expression patterns used for isolation of parasegment nuclei are shown, visualized with antibody to the FLAG epitope on the INTACT fusion protein. Each panel shows embryos at about 6, 8, 10, 12, 14, and 16 hr after fertilization, with the stained cells marking the indicated parasegments. Embryos between 5 hr and 13 hr old (or between 4 hr and 10 hr for PS6) were harvested for analysis.

*Figure 1. Continued on next page*

shows a PS4 anterior border (*Casares et al., 1997*), and a ncRNA with a PS4 anterior expression limit is transcribed from the region 5–15 kb downstream of the *Ubx* start (*Pease et al., 2013*). Despite the activity of these regulatory regions in PS4 cells, we did not detect a decrease in K27me3 at these locations. The PS4 active state may be early and transient, limited to a subset of PS4 cells, or resistant to H3K27 methylation.

We have profiled additional histone modifications and several chromosomal proteins in single parasegments to understand better how domains are established and bounded. The sharp discontinuities in K27 methylation coincide with binding sites of the CCCTC-binding factor (CTCF), which is thought to impose boundaries on regulatory domains in many systems (*Ohlsson et al., 2010*). *Figure 2G* shows the CTCF ChIP pattern in unsorted nuclei, similar to that previously reported (*Nègre et al., 2010*). CTCF in PS7 nuclei gives a pattern virtually identical to the whole embryo pattern (*Figure 2H*). Apparently, the activation state of a domain is not coupled to the presence of CTCF at its borders. We note that our experimental conditions do not detect robust CTCF enrichment in some BX-C locations that have been reported by others (*Nègre et al., 2010*), notably at the border between regulatory regions for PS7 and PS8, and proximal to the 3′ end of the *Ubx* transcription unit.

CTCF is present in a protein complex with Centrosomal Protein 190 (CP190) (*Mohan et al., 2007*), and the binding sites of these proteins often overlap (*Nègre et al., 2010*). The CP190 sites largely coincide with CTCF sites in the BX-C, and, as with CTCF, there is little difference between the CP190 pattern in unsorted nuclei (*Figure 2I*; *Nègre et al., 2010*) and in PS7 nuclei (*Figure 2J*). There is a noteworthy CP190 site immediately proximal to the 3′ end of the *Ubx* transcription unit, separating *Ubx* from the adjacent gene (*modular serine protease*). In the Antennapedia complex, the other homeotic gene complex in flies, CP190 peaks often appear without coincident CTCF peaks (*Figure 2—figure supplement 2*).

Acetylation of H3K27 (H3K27ac) is correlated with active enhancer regions (*Heintzman et al., 2009*; *Kharchenko et al., 2011*), and is anticorrelated with PcG function (*Tie et al., 2009*), prompting us to examine its pattern in the BX-C. H3K27ac showed the opposite pattern to H3K27me3. There is relatively little H3K27ac across the BX-C in unsorted nuclei, but in PS7, the active domains show broad enrichment for H3K27ac (*Figure 3B,C*). The H3K27ac levels differ

*Figure 1. Continued*

The following figure supplements are available for figure 1:

**Figure supplement 1**. Tools for parasegment-specific ChIP-seq.

**Figure supplement 2**. Close-up of parasegment-specific expression patterns.

in the three active domains, for reasons we do not yet understand. Within each domain, however, the coverage is boundary-to-boundary, although many enhancers in the BX-C are not active in embryos. Overall, H3K27ac is a mirror to H3K27me3, in extent, if not in density.

ChIP of serine-5-phosphorylated RNA polymerase II (Pol-II) (transcriptionally engaged polymerase) in whole embryos shows the transcription units for *Ubx*, *abd-A*, and *Abd-B* (*Figure 3D*). Embryonic non-coding transcripts (*Bae et al., 2002*; *Pease et al., 2013*) are not apparent from the Pol-II density, either because they are limited to very early stages of embryogenesis, or because the levels of transcription are too low. There is an unexpected Pol-II peak ~14.5 kb upstream of the *Ubx* promoter, coincident with the CTCF binding site at the border between the PS5 and PS6 domains (arrow in *Figure 3D*). There is no reported transcript initiating at this position (*Pease et al., 2013*), although a short transcript could have been missed. Pol-II in PS7 (*Figure 3E*) marks the *Ubx* and *abd-A* transcription units, as expected from the UBX and ABD-A protein patterns (*Figure 1—figure supplement 2*). There is also a Pol-II peak at the major *Abd-B* promoter (arrow in *Figure 3E*), although *Abd-B* RNA and protein are absent from PS7 (*Celniker et al., 1990*; *Boulet et al., 1991*) and this region is H3K27 trimethylated. Analogous localization of Pol-II at the *Abd-B* promoter in wing and haltere discs was reported by *Chopra et al. (2009)*, who argued that these polymerase molecules were 'paused'. The H3K4me3 mark aligns with the *Ubx* and *abd-A* promoters in PS7 nuclei (but not *Abd-B*; *Figure 3G*) and is reduced in mixed cells (*Figure 3F*); this histone mark has been seen at active promoters in other systems (*Ruthenburg et al., 2007*).

The POLYCOMB subunit of PRC1 is known to bind to the H3K27me3 modification through its chromodomain (*Fischle et al., 2003*), and so PRC1 localization might be expected to correspond to the stair-step H3K27me3 pattern. Published ChIP–CHIP and ChIP-seq profiles for PC across the BX-C vary from broad coverage matching the H3K27me3 profile (*Kwong et al., 2008*) to sharp peaks at the known PREs (*Gonzalez et al., 2014*). The difference may be due to different procedures for chromatin fragmentation prior to immunoprecipitation (*Straub et al., 2013*). Our PC profiles show sharp peaks at PREs in nuclei from whole embryos (*Figure 3H*), matching the profile for POLYHOMEOTIC, another PRC1 subunit (*Figure 3J*). In PS7 nuclei, we find both the POLYCOMB and POLYHOMEOTIC subunits of PRC1 are bound to the Polycomb Response Elements of the active PS6 and PS7 domains at levels comparable to those in the whole embryo profile (*Figure 3H–K*). In the active PS5 domain, the levels of PC and PHO at the PS5 PRE are reduced, relative to the whole embryo profile. *Papp and Müller (2006)* have reported reduced retention of PRC1 and 2 components at the PRE in the PS5 domain in haltere and third leg imaginal discs, where the PS5 domain is in the active state and K27me3 is absent. *Kwong et al. (2008)* have made analogous observations regarding PRC retention in imaginal discs of the third thoracic segment. PRC1 presence is apparently not a determinant of repressed domains, although its subunits are clearly needed for PcG repression (*Simon and Kingston, 2013*). It will be important to map H2A ubiquitylation (a function of the dRING subunit of PRC1) across the domains, especially given recent findings that H2A ubiquitylation promotes PRC2 recruitment and H3K27 trimethylation (*Blackledge et al., 2014*; *Cooper et al., 2014*; *Kalb et al., 2014*).

The profiles for SU(Z)12, a PRC2 component (*Figure 3L,M*), look similar to those of the PRC1 proteins, POLYCOMB and POLYHOMEOTIC. There are major peaks at the known PREs in both whole embryo and PS7 nuclei, although in PS7, the PREs in active domains have reduced levels of SU(Z)12. This result is again consistent with the prior work of *Papp and Müller (2006)* on the PS5 PRE in imaginal discs. SU(Z)12 is also present at the transcription start sites of *Ubx*, *abd-A*, and *Abd-B*; binding at these promoters is more apparent than it is for PRC1 proteins.

## Discussion

The Polycomb Group repression system is often described as a cellular memory mechanism, which can impose lifelong silencing of a gene in response to a transitory signal. That view seems valid, but the concept of a PcG regulatory domain is much richer. In the PS6 domain of the BX-C, for example, there are many enhancers to drive *Ubx* expression in specific cells at specific developmental times, all of which are blocked in parasegments one through five, but active in parasegments 6 through 12

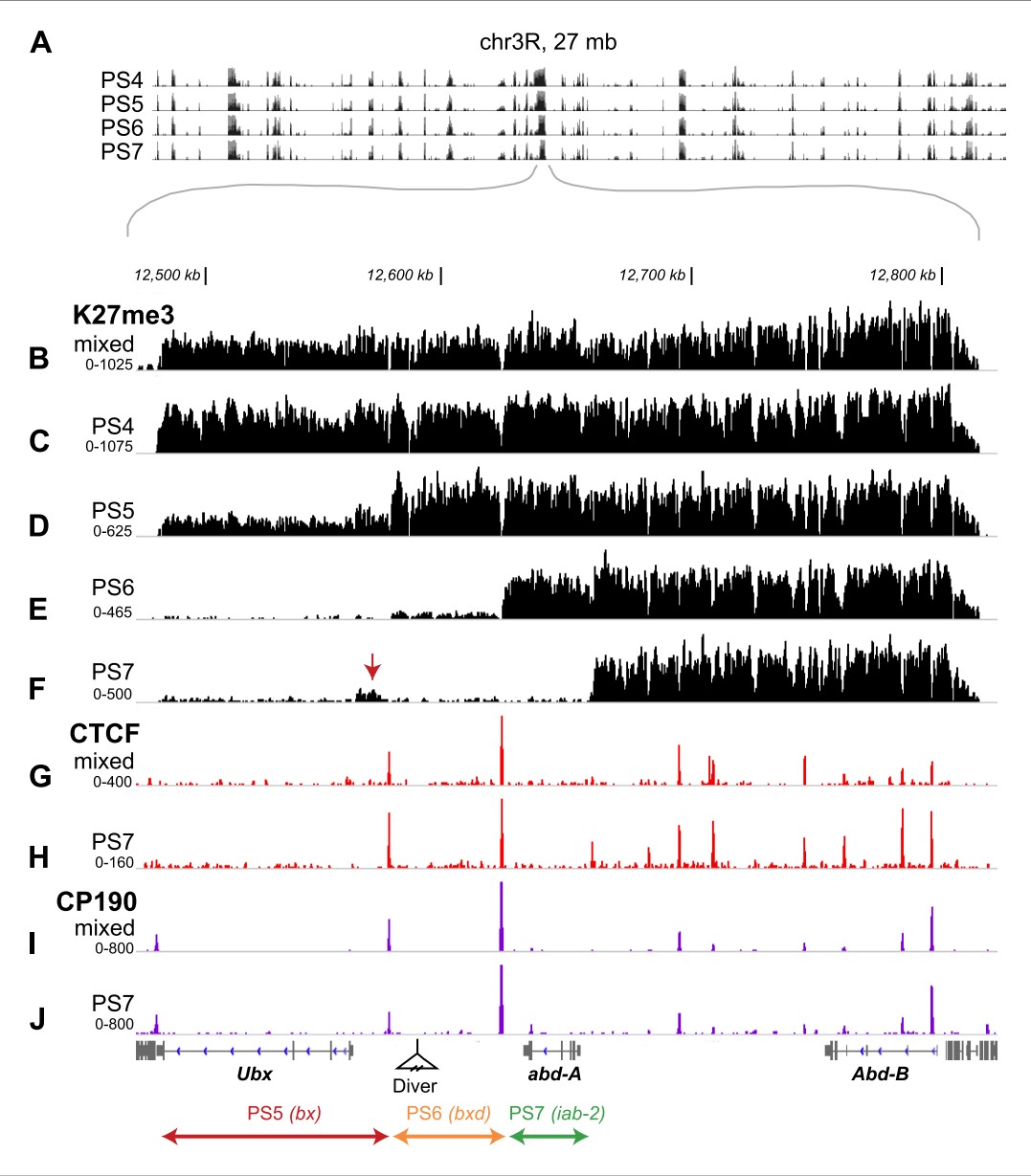

**Figure 2**. H3K27me3, CTCF, and CP190 profiles. (**A**) ChIP-seq profiles across 27 Mb of chromosome 3R are virtually identical, except at the BX-C, in the middle of the chromosome arm. (**B–F**) H3K27me3 profiles across 380 kb encompassing the BX-C. (**G–H**) CTCF binding sites. (**I** and **J**) CP190 binding sites. Panels **B**, **G**, and **I** were prepared from unsorted nuclei; parasegment-specific nuclei were used for the other panels, as indicated. Transcription units of coding genes are shown below the profiles. The Drosophila reference sequence includes a 6.1 kb Diver retroposon insertion at the indicated position; it was not present in the strains used for this analysis. At the bottom are shown the three regulatory domains defined by this analysis.

The following figure supplements are available for figure 2:

**Figure supplement 1**. Genome-wide comparisons of H3K27me3 patterns.

**Figure supplement 2**. H3K27me3, CTCF, and CP190 profiles in PS7 for the Antennapedia complex, illustrated as in *Figure 2*.

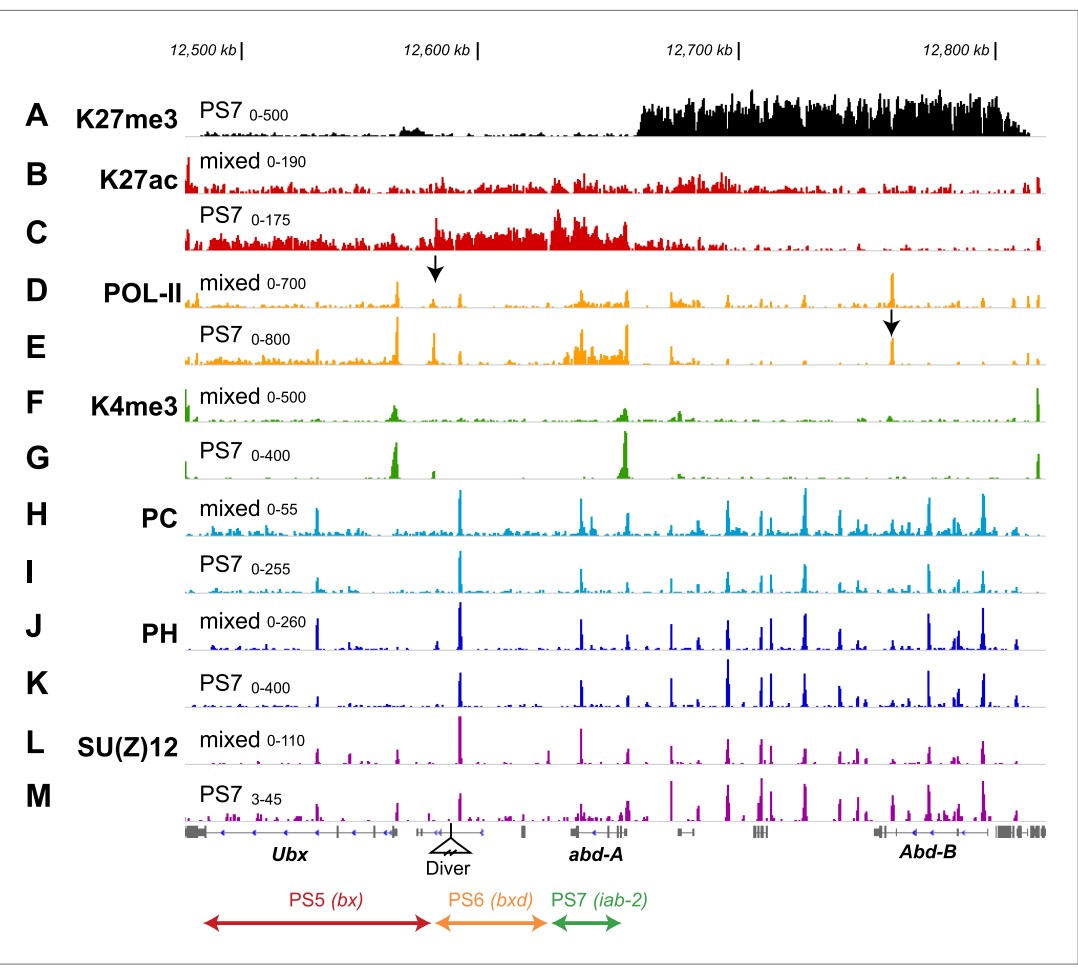

**Figure 3**. Additional features of the BX-C in PS7. (**A**) The H3K27me3 distribution, repeated from **Figure 2F**, marks the border between active and repressed domains. (**B** and **C**) H3K27ac profiles are shown for mixed and PS7 nuclei; the acetylation mark in PS7 is largely restricted to the active domains. (**D** and **E**) Pol-II profiles show peaks at the promoters of *Ubx*, *abd-A*, and *Abd-B* both in whole embryo and in PS7 nuclei. In PS7, polymerase is distributed across the transcription units of *Ubx* and *abd-A*, but not *Abd-B*. The arrow in **D** marks the PS5/PS6 border; the arrow in **E** marks the major *Abd-B* promoter. (**F** and **G**) H3K4me3 profiles show prominent peaks over the *Ubx* and *abd-A* promoters, both of which drive strong transcription in PS7. (**H** and **I**) POLYCOMB profiles show sharp peaks over all known Polycomb Response Elements. In PS7, there is a reduction of the PC peak at the 'bx PRE' in the active PS5 domain. There is also a PC peak at the *abd-A* promoter, which is somewhat reduced in PS7, where *Abd-A* is transcribed. (**J** and **K**) POLYHOMEOTIC protein shows peaks at known PREs and the *abd-A* promoter in both the active and repressed domains. As with PC, the PH peak at the bx PRE is reduced in PS7. (**L** and **M**) SU(Z)12 profiles also mark PREs, but in PS7, the PRE peaks are reduced in all three active domains. The ranges for the vertical axes are indicated above the left edge of each trace.

(**Bender and Lucas, 2013**). Individual enhancers need not include a segmental address that is specified, for example, by gap and pair-rule DNA-binding factors; their function is segmentally restricted by the domain architecture. Indeed, these enhancers will drive expression in a different parasegment when inserted into a different domain (as in the *Cbx* transposition, **Peifer et al., 1987**). Each domain has a distinctive collection of enhancers; the UBX pattern in PS5 is quite different from that in PS6. Thus, there are two developmental programs for *Ubx*, one in each of these parasegments, without the need for a duplication of the *Ubx* gene. Other loci with broad regions of H3K27 methylation may likewise be parsed into multiple domains, once we examine histone marks in specific cell types.

The all-or-nothing H3K27me3 coverage of the BX-C parasegmental domains validates and refines the domain model. In particular, K27me3 is uniformly removed across the PS5 and PS7 domains in PS5 and PS7, even though the activated genes in those parasegments (*Ubx* and *abd-A*, respectively) are

only transcribed in a subset of cells. It is interesting that both PRC1 and PRC2 components have binding patterns that do not fully reflect function (repression and K27 methylation, respectively), indicating the possibility that function of these complexes is regulated separately from binding (*Papp and Müller, 2006*). The challenges now are to understand how PcG regulated domains are established, differently in different parasegments, and to describe the molecular mechanisms, including changes in chromosome structure, that block gene activity in H3K27 trimethylated domains.

## Materials and methods

### Gal4 and Gal80 sources

A P element was built containing both Gal4 and Gal80 (*Figure 1—figure supplement 1A*). The P promoter initially drives Gal4 transcription. The P-element used the Gal4 gene with an HSP70 terminator derived from pGawB (*Brand and Perrimon, 1993*), fused in frame to the P transposase gene after the 130th amino acid. The Gal4 gene was followed by a synthetic FRT site plus a polylinker. The Gal80 coding sequence with an HSP70 terminator, derived from pBPG80Uw (a gift of B Pfeiffer and G Rubin) was inserted into this polylinker. Another synthetic FRT site was inserted near the start of the Gal4 coding region at a unique *Sph*I site. Thus, the Gal4 coding sequence is flanked by FRT sites, so that flipase-induced recombination can turn the P element into a Gal80 producer, with the same expression pattern. The P element was transformed into a random chromosomal position, and then used to swap into the position of an existing P enhancer trap in the BX-C (*Bender and Hudson, 2000*) or at the *Antennapedia* promoter. P element swapping (*Sepp and Auld, 1999*) is a low frequency event, but the lines were constructed with UAS/GFP as a marker for Gal4 activity, and thousands of first instar larvae could be quickly screened for one with the expected segmentally-restricted GFP pattern. A derivative of this vector was made by replacing the HSP70 terminator downstream of the Gal80 coding sequence with the poly(A) addition site from the *Drosophila Alcohol Dehydrogenase* (*Adh*) gene, recovered as a 317 bp PCR fragment from genomic DNA.

We have also taken known embryonic enhancers from the BX-C and inserted them into a second vector (*Figure 1—figure supplement 1B*), which was derived from the initial vector with the *Adh* terminator. At a *Bam*H1 site immediately upstream of the 5′ FRT site, a fragment was inserted containing: (1) an 841 bp fragment from the 5′ end of the *gypsy* mobile element (bases 189-1029), (2) an *Asc*I site, and (3) a fragment from the *engrailed* gene (−595 bp to +243 bp relative to the major transcription start site). The *gypsy* fragment includes twelve tandem binding sites for SUPPRESSOR OF HAIRY WING (SU(HW)). The *engrailed* fragment includes the proximal PRE ('PSE2' [*DeVido et al., 2008*]), and is designed to fuse the first nine amino acids of ENGRAILED to the amino terminal end of Gal4. Enhancer fragments from the BX-C were cloned by PCR with *Asc*I extensions on the primers, and inserted into the *Asc*I site between the *gypsy* and *engrailed* fragments. The engrailed PRE was included to maintain the segmentally-restricted pattern through most of embryonic development, and the SU(HW) binding sequences were intended to insulate the promoter from endogenous enhancers near the insertion site. Transformants were recognized by Gal4-driven GFP patterns in young larvae, and, again, any Gal4 producer could be converted to a Gal80 producer. Sequences of these P elements are available on request.

### Swap positions

P element swapping was used to replace a P element at the *Antennapedia* P1 promoter (P[XP] Antpd02480, target site duplication: 3R: 2,825,198-2,825,205 [*Thibault et al., 2004*]), and a P element in the PS8 (iab-3) domain of the BX-C (HCJ192, target site duplication 3R: 12,673,279-12,673,286 [*Bender and Hudson, 2000*]). Other swaps were made with P elements in the PS5 (bx) and PS6 (bxd) domains, but these did not give sufficiently strong or uniform expression patterns. The *Antp* swap was subjected to P transposase to induce 'local hopping' and thereby increase the Gal4 expression level.

### Enhancer fragments

Fragments used as enhancers inserted into the vector of *Figure 1—figure supplement 1B* included:

    abx enhancer (for PS5 anterior limit) 3R: 12,508,951-12,513,096
    pbx enhancer (for PS6 anterior limit) 3R: 12,598,546-12,600,177
    iab-2 enhancer (for PS7 anterior limit) 3R: 12,636,236-12,639,140
    iab-3 enhancer (for PS8 anterior limit) 3R: 12,664,301-12,666,886

Other fragments tested covered the 'bx' and 'bxd' enhancers, but these did not give sufficiently strong or uniform expression patterns. One insertion of a P element with the iab-3 enhancer fortuitously inserted 1.4 kb upstream of the *Ubx* transcription start site (target site duplication, 3R: 12,561,577-12,561,584). This gave a strong pattern with a sharp PS6 anterior limit, and was used as the Gal4 source for PS6 nuclei. Insertions with the abx and pbx enhancers were subjected to P transposase to adjust the levels of Gal4 and Gal80. *D. melanogaster* genome coordinates are from release 5.57.

## INTACT marker

Flies with the INTACT reporter in a P element on the third chromosome (w[1118]; p[w+; UASRG]6) were generously provided by Paul Talbert and Steve Henikoff. This P element was crossed to a P transposase source (P{ry[+t7.2] = Delta2-3}99B), and offspring were screened initially for darker eye color, most likely due to multiple copies of the INTACT element. Such stocks with INTACT insertions on the second or third chromosome were then crossed to a Gal4 source, and those with brightest mCherry expression were used for subsequent efforts to mark single parasegments. The higher copy INTACT chromosomes are designated 'INTACT$_n$ on II' and 'INTACT$_n$ on III'.

## Genotypes

To mark PS4, males homozygous for [w; abx en.>Gal80, pbx en.>Gal80; Antp swap>Gal4*, INTACT on III] were crossed to females homozygous for [w; INTACT$_n$ on II; INTACT$_n$ on III].

To mark PS5, males homozygous for [w; abx en.>Gal4*, pbx en.>Gal80*, UAS-GFP*] were crossed to females homozygous for [w; INTACT$_n$ on II; INTACT$_n$ on III].

To mark PS6, males homozygous for [w; INTACT$_n$ on II; iab-2 en.>Gal80, iab-3 swap>Gal80, INTACT on III] were crossed to females [w; bxd insert>Gal4, iab-2 en.>Gal80, iab-3 en.>Gal80, INTACT on III / TM6]. Only half of the resulting embryos had a Gal4-driven stripe.

To mark PS7, males homozygous for [w; iab-2 en.>Gal4, iab-3 swap>Gal80, INTACT on III] were crossed to females homozygous for [w; INTACT$_n$ on II; INTACT$_n$ on III].

Insertions treated with P transposase are marked above with an asterisk (*).

## Embryo collection and nuclear preps

Agar plates with yeast paste were put on the PS4, PS5 or PS7 laying cages for 8 hr at 25°C (or 16 hr at 18°C), and then the eggs were aged for another 5 hr at 25°C (or 10 hr at 18°C). For PS6, a 6 hr collection was aged for a further 4 hr. The eggs were dechorionated (2 min in 50% bleach) and fixed in fixation buffer (5% formaldehyde, 100 mM NaCl, 50 mM Hepes, 1 mM EDTA, 0.5 mM EGTA, pH 8.0) under heptane, on a rotator for 15 min at 25°C. The fixed embryos were washed with stop buffer (125 mM glycine, 130 mM NaCl, 7 mM Na$_2$HPO$_4$, 3 mM KH$_2$PO$_4$, 0.1% Triton X100, pH 8.0) for 2 min at 25°C, and then briefly with PBS (130 mM NaCl, 7 mM Na$_2$HPO$_4$, 3 mM KH$_2$PO$_4$) plus 0.1% Triton X100. The washed embryos were collected on a nytex filter, and examined under an epifluorescence stereoscope. Embryos that appeared too old, with fluorescent nuclei anterior to the desired parasegment, were manually removed. The remaining eggs were collected and weighed in a 1.5 ml microfuge tube, flash frozen in liquid nitrogen, and stored at −75°C.

For the isolation of nuclei, ~100 mg of embryos were suspended in 5 ml of BBT buffer (55 mM NaCl, 40 mM KCl, 15 mM MgSO$_4$, 5 mM CaCl$_2$, 10 mM Tricine, 20 mM dextrose, 50 mM sucrose, 0.1% bovine serum albumin (Miles PENTEX), 0.01% Triton X-100, pH7.0) plus 100 µl protease inhibitor cocktail (Roche Complete, EDTA-free), and disrupted with 10 strokes in a Dounce homogenizer with a loose pestle, then 10 strokes with a tight pestle. The homogenate was spun at 275 × g for 1 min to remove large debris. The resulting supernatant was spun at 1000 × g for 10 min to pellet the nuclei. The pellet was resuspended in 2 ml BBT buffer plus 40 µl protease inhibitor, and homogenized again, 20 strokes with the tight pestle. The homogenate was passed through a cell strainer cap with a 40 µm nylon mesh and stored at 0°C. Hoechst 33,342 dye was added (2 µl of 10 mg/ml solution) before sorting.

## Sorting

Nuclei were sorted on a BD Biosciences (San Jose, CA) FACSAria IIu instrument. The fluorescence from Hoechst 33342 (405 nM excitation, 425–475 nm emission) was used to select for a 2N DNA content, thus avoiding clumps of nuclei. These single nuclei were secondarily selected for mCherry fluorescence (594 nM excitation, 619–640 nM emission) stronger than 99.9% of nuclei from control (*Oregon R*) embryos.

Recovered nuclei constituted between 1 and 6% of total single nuclei (*Figure 1—figure supplement 1C*), depending of the parasegment under selection. Recovered nuclei were kept at 0°C, and used for chromatin preparation within 5 hr.

## Chromatin preparation

The yield of sorted nuclei in sort buffer (PBS) was determined with a hemacytometer, and 50–200 K nuclei were used for each ChIP. Chromatin fragmentation was accomplished either with micrococcal nuclease (MNase) or bath sonication. For MNase fragmentation, nuclei in sort buffer were supplemented with PBS containing 0.1% Triton-X100 (PBS-Tx) to a volume of 400 µl, and CaCl2 to 1 mM. Nuclei were pre-warmed to 37°C and digested with 12U MNase (Worthington Biochemical) for 3 min. Digestion was stopped by moving the tubes to ice and adding 10 µl of 250 mM EDTA, 250 mM EGTA, and nuclei were briefly sonicated in a Diagenode Bioruptor (Denville, NJ) (3 min, high intensity, 30 s on, 30 s off). For bath sonication, the nuclear suspension in sort buffer was adjusted to ChIP buffer conditions (10 mM Tris pH8, 100 mM NaCl, 0.1% sodium deoxycholate, 0.5% sarkosyl, 1% Triton-X100) in a volume of 500 µl. Nuclei were sonicated for 30 min at high intensity (30 s on, 30 s off) in a Bioruptor. Fragmented chromatin was snap frozen and stored at −80°C prior to ChIP.

## ChIP-seq

MNase fragmented chromatin was adjusted to ChIP buffer conditions in a final volume of 500 µl, and incubated on ice for 5 min. Prior to ChIP, protease inhibitors (COMPLETE, Roche) were added to either MNase fragmented chromatin or sonicated chromatin. Chromatin was subjected to a high speed spin at 4°C for 10 min. Supernatant was moved to a new low retention tube, 1% input was removed and stored at 4°C, and and the remaining chromatin was incubated overnight with the appropriate antibody at 4°C. After a high speed spin at 4°C for 10 min, supernatant was moved to a low retention tube with 10 µl prewashed protein A dynabeads (Life Technologies), and incubated at 4°C for 1–2 hr with gentle rotation. After six rinses with ice cold ChIP buffer, immunoprecipitates were eluted from the beads with two successive additions of 125 µl freshly made elution buffer (0.2% SDS, 0.1 M NaHCO3, 5 mM DTT) incubated at 65°C for 10 min. Eluates were combined, and adjusted with Tris and EDTA to a final concentration of 10 mM Tris pH8 and 2 mM EDTA. 250 µl elution buffer was added to the reserved input and likewise adjusted. DNA cleanup began by incubating with RNase (DNase-free, Roche) at 37°C for 30 min, proteinase K (PCR-grade, Roche) at 55°C for 1 hr, and crosslinks were reversed by incubating at 65°C for 1 hr. After two phenol-chloroform extractions and one chloroform extraction, purified DNA was ethanol precipitated in the presence of sodium acetate and Glycoblue (Ambion). Sequencing library construction was performed according to (*Bowman et al., 2013*). Libraries were sequenced on an Illumina (San Diego, CA) HiSeq2000, HiSeq2500, or MiSeq according to manufacturer's instructions.

## ChIP antisera

anti-K27me3 (Active Motif 39155), anti-K27ac (Active Motif 39,136), anti-K4me3 (07473; Millipore), anti-CTCF (*Smith et al., 2009*), anti-Ph (residues 772–984 [*Oktaba et al., 2008*]), anti-CP190 (*Pai et al., 2004*), anti-Pol-II (ab5131; Abcam), anti-Pc (*Schuettengruber et al., 2009*), anti-SU(Z)12 (*Müller et al., 2002*).

## Data analysis

ChIP-seq reads were aligned to the dm3 genome using BWA and allowing up to one mismatch. Potential PCR duplicates, tags with multiple alignments, and paired-end reads with insert size greater than 1 kb were removed from further analysis. We calculated normalized positional coverage as previously described (*Pinter et al., 2012*; *Yildirim et al., 2012*). Briefly, experimental coverage was normalized by corresponding input coverage for each position: $n_{norm} = [(n+1)/(n_i+1)] * [N_i/N]$, where $n$, $n_i$, $N$, and $N_i$ are positional coverages in experiment and input, and total genome coverages in experiment and input, respectively.

To determine regions of enrichment, we summed the coverage within sliding windows of size 1 kb and step 50 bp across the entire genome. The same was calculated for tags assigned a random location within the chromosome. The resulting distribution of random window coverages was then used to determine the significance of observed window coverages. Individual p-value cutoffs were selected based on manual inspection of each experiment. For samples with replicates, only regions significant in both replicates were used.

For genome-wide analysis of coverage difference between two parasegments for H3K27me3, we first merged the significant regions of enrichment from the different parasegments to create an aggregate set of regions of interest. Regions greater than 50 kb were split into equally sized

sub-regions. For each region we calculated the density, or total normalized coverage divided by region length. Replicate density values were averaged. To find differences, region densities for two parasegments were scatter plotted against one another, and a line of best fit calculated. Points located furthest away from this fit line were identified as the candidate differing regions. This approach allows us to account for the variation in sequencing coverage and noise for each sample and parasegment.

SPP (*Kharchenko et al., 2008*) was used to generate tag density profiles for visualization. For each chip ChIP experiment, we followed standard methods to estimate the binding peak separation distance, remove low quality or anomalous tags, and generate an input-subtracted, Gaussian-smoothed tag density profile.

## Acknowledgements

The datasets supporting the results of this report are available in NCBI's Gene Expression Omnibus and are accessible through GEO Series accession number GSE55257. We are grateful to Barret Pfeiffer for the pBPG80Uw plasmid containing a codon-optimized version of Gal80. Paul Talbert and Steve Henikoff supplied us with a UAS/INTACT P element. Stephanie Terrizzi and Jodi Moore assisted with nuclear sorting. PH and SU(Z)12 antiserum were a gift from Jürg Müller, and CTCF antiserum was a gift from Victor Lubanenkov. Matt Simon, Michele Markstein, and Jesse Cochrane gave helpful comments on the manuscript. SKB was supported by the Damon Runyon Cancer Research Foundation. This work was supported by grants from the NIH, GM43901 to REK and GM28630 to WB.

## Additional information

### Funding

| Funder | Grant reference number | Author |
| --- | --- | --- |
| National Institutes of Health | GM43901 | Robert E Kingston |
| National Institutes of Health | GM28630 | Welcome Bender |
| Damon Runyon Cancer Research Foundation | 1995-08 | Sarah K Bowman |

The funders had no role in study design, data collection and interpretation, or the decision to submit the work for publication.

### Author contributions

SKB, AMD, WB, Conception and design, Acquisition of data, Analysis and interpretation of data, Drafting and revising the article; HD, Conception and design, Acquisition of data; PIW, RIS, Analysis and interpretation of data, Drafting and revising the article; REK, Conception and design, Analysis and interpretation of data, Drafting and revising the article

## Additional files

### Major dataset

The following dataset was generated:

| Author(s) | Year | Dataset title | Dataset ID and/or URL | Database, license, and accessibility information |
| --- | --- | --- | --- | --- |
| Bowman SK, Deaton AM, Domingues H, Wang PI, Sadreyev RI, Kingston RE, Bender W | 2014 | H3K27 modifications define segmental regulatory domains in the *Drosophila* bithorax complex | http://www.ncbi.nlm.nih.gov/geo/query/acc.cgi?acc=GSE55257 | Publicly available at NCBI Gene Expression Omnibus. |

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
