## [Decision Letter]

[Editors’ note: although it is not typical of the review process at *eLife*, in this case the editors decided to include the reviews in their entirety for the authors’ consideration as they prepared their revised submission.]

Thank you for sending your work entitled “H3K27 modifications define segmental regulatory domains in the Drosophila bithorax complex” for consideration at *eLife*. Your article has been favorably evaluated by K VijayRaghavan (Senior editor) and 3 reviewers, one of whom is a member of our Board of Reviewing Editors.

The Reviewing editor and the other reviewers discussed their comments before we reached this decision, and the Reviewing editor has assembled the following comments to help you prepare a revised submission.

We think that the study is solid and has its value. However, we collectively believe that the current data set (H3-K27me3/ PH/ CTCF) is too preliminary compared to earlier studies but that it might become suitable if the analysis is expanded as suggested below.

The authors should look minimally at PRC2 binding and RNA Pol II occupancy. These two profiles would help to clarify whether the 'constitutive' binding of PRC2, PRC1 etc. at the HOX gene *Ubx* (i.e. in *Ubx* OFF and *Ubx* ON cells) that was reported back in 2006 is a general principle. Similarly, it was shown that several components of the general transcription machinery are bound at *Ubx* in the OFF state, and so it would be good to know whether assembly of the RNA Pol II machinery even at permanently repressed Polycomb target genes is a general principle.

We also noticed that the citation of relevant scientific literature is missing.

*Reviewer #1*:

Figure 1:

Can the authors computationally overlay the different parasegments that are labeled at each time point? Currently, it is visually difficult to assess the level of overlap or lack thereof between each of the labeled parasegments.

Is it possible to label any upstream parasegments? The question of contamination of PS4 by upstream signal may thereby be resolved.

The authors should provide antibody stainings of genes within the BX-C cluster for us to align against the labeled parasegments.

Figure 2:

What about the Antp-C locus? Does H3K27me3 profiling parallel what is seen in the BX-C locus? Does dCTCF localize to boundaries here as well?

As there are many characterized insulator proteins in the fruit fly, how does dCTCF co-localize with these factors? I.e. CP190, BEAF-32, Su(Hw)?

How is a given constitutive CTCF site chosen to be “active” in performing a boundary function? For example, in PS7, why are activation signals able to push through the most anterior dCTCF sites? Is specificity achieved through association with distinct insulator partners at different CTCF sites?

How does the H3K27me3 pattern compare to the distribution of E(Z) or other PRC2 components? Does PRC2 only sit at PREs, from which the modification spreads? Is PRC2 absent from upstream PREs present within an activated region of BX-C? If present, what inhibits PRC2 activity at these sites?

Since the authors are able to purify nuclei from distinct parasegments of the fly, they should be able to provide RNA-seq or Pol-II occupancy tracks to correlate with presence of repressive H3K27me3 and active H3K27ac.

The genetic system employed nicely allows for the isolation of parasegments during time points of development. What are the dynamics of the activation of the BX-C locus? Is the transition gradual or immediate? Is it possible to ChIP some of the transcription factors thought to activate these genes?

What is the 3D conformation of the parasegments within the BX-C cluster? How do they compare to each other? Are active and repressive domains spatially partitioned?

How is the expression and H3K27 distribution altered in mutants (null or hypomorphs) of Polycomb genes? Is there simply a decrease in H3K27me3, or is the H3K27me3 boundary shifted as well?

Figure 3:

How does the distribution of H2AK119Ub correlate with Polyhomeotic? Does it form blocks within transcriptionally repressed chromatin, as is the case for PRC2?

*Reviewer #3*:

1 )The authors should analyze the PRC2 binding profile (e.g. ChIP with antibody against E(z))in PS4, 5, 6 and 7, or at the minimum, just in PS7 as they did for Ph, CTCF etc.. This would address the conceptually important point whether the binding of PRC2 at PREs is constitutive (i.e. PRC2 bound not only in the inactive but also in the active state; see also below) and that the control takes place at the level of deposition of the modification. Given that the authors may already have chromatin from PS7 nuclei or can generate it as they did for several of their experiments in the manuscript, I do not think that it is asking too much to perform such a ChIP experiment.

2) This point is directly connected to the first point. The authors must discuss their findings in the context of the findings from earlier studies (i.e. [35]; Kwong et al., PLOS Genetics 2008; Soshnikova and Duboule, Science 2009). Directly relevant to the work by Bowman here, Papp and Muller had compared the chromatin of the HOX gene *Ubx* in wing imaginal disc cells where *Ubx* is inactive and in haltere and third leg discs where the gene is active. One key conclusion from that work was that PRC2 (as well as PRC1 and PhoRC) is bound at *Ubx* both in the inactive and in the active state (with subtle differences) but that the domains of H3-K27me3 are different in the two states, with H3-K27me3 covering the entire upstream, promoter and coding region in wing disc cells but being absent from the promoter and coding region in haltere and third leg disc cells. So the observation that H3-K27me3 domains correlate with inactivity of regulatory regions in the bithorax complex is not new but the nice data shown here in embryos (with modern genomic techniques) extend and refine these findings. From the above it also directly follows that it is key to find out about PRC2 binding at the *bx*, *bxd* and *abd-A* PREs in PS7 (Point 1).

---

## [Author Response]

Reviewer #1:

Figure 1*:*

Can the authors computationally overlay the different parasegments that are labeled at each time point? Currently, it is visually difficult to assess the level of overlap or lack thereof between each of the labeled parasegments.

The authors should provide antibody stainings of genes within the BX-C cluster for us to align against the labeled parasegments.

**Response:** New figure with side-by-side embryos from each strain marking single parasegments, and a panel with UBX and ABD-A stainings.

**Location in revised manuscript:**
Figure 1—figure supplement 2

This new figure should clarify the position of the labeled parasegments relative to each other and to UBX and ABD-A expression.

*Is it possible to label any upstream parasegments? The question of contamination of PS4 by upstream signal may thereby be resolved*.

Construction of these strains takes many months. In any case, labeling more anterior parasegments would not resolve whether the PS4 (or PS5) nuclei are contaminated by anterior cells.

Figure 2*:*

*What about the Antp-C locus? Does H3K27me3 profiling parallel what is seen in the BX-C locus? Does dCTCF localize to boundaries here as well*?

**Response:** New figure depicting H3K27me3, CTCF, and CP190 in the Antp-C.

**Location in the revised manuscript:**
Figure 2—figure supplement 2.

H3K27me3 is depleted over the Antp transcription unit, as expected. CTCF is less prevalent in the Antp-C than in the BX-C. In the Antp-C, CP190 predominates.

*As there are many characterized insulator proteins in the fruit fly, how does dCTCF co-localize with these factors? I.e. CP190, BEAF-32, Su(Hw)*?

**Experimental response:** ChIP-seq for CP190 in mixed cells and parasegment 7.

**Location in revised manuscript:**
Figure 2

Since CP190 is in a protein complex with CTCF and can alter local chromatin structure in a way that may promote boundary function, we thought it would be a worthwhile target of investigation. It co-occupies many sites with CTCF, but also binds to the BX-C and the ANTP-C at locations where we do not detect CTCF.

*How is a given constitutive CTCF site chosen to be “active” in performing a boundary function? For example, in PS7, why are activation signals able to push through the most anterior dCTCF sites? Is specificity achieved through association with distinct insulator partners at different CTCF sites*?

Our results suggest that boundaries are constitutive. They may have nothing to do with whether an adjacent domain is active or repressed; they may only prevent the spread of activation or repression. The comparison of CTCF and CP190 profiles suggests that each boundary may be distinctive, which is borne out by sequence comparisons and swapping experiments from other labs, but this is no reason to suggest that the components of any boundary vary from one parasegment to another.

*How does the H3K27me3 pattern compare to the distribution of E(Z) or other PRC2 components? Does PRC2 only sit at PREs, from which the modification spreads? Is PRC2 absent from upstream PREs present within an activated region of BX-C? If present, what inhibits PRC2 activity at these sites*?

**Experimental response:** ChIP-seq for Su(z)12 in mixed cells and parasegment 7.

**Location in revised manuscript:**
Figure 3

In mixed cells, Su(z)12 is present in sharp peaks at PREs and transcription start sites. In PS7, Su(z)12 peaks are somewhat diminished at the PREs in the active domains, while the peaks at the transcription start sites are strengthened. The reasons for these subtle changes will require further investigation.

Since the authors are able to purify nuclei from distinct parasegments of the fly, they should be able to provide RNA-seq or Pol-II occupancy tracks to correlate with presence of repressive H3K27me3 and active H3K27ac.

**Experimental response:** ChIP-seq for Pol-II in mixed cells and parasegment 7.

**Location in revised manuscript:**
Figure 3

Transcriptionally engaged Pol-II is present at the promoter and throughout gene bodies of *Ubx* and *Abd-A* in PS7, consistent with the expression of these genes. Pol-II is also enriched at the major promoter of *Abd-B*, even though this gene is transcriptionally silent in PS7 and covered by H3K27me3.

*The genetic system employed nicely allows for the isolation of parasegments during time points of development. What are the dynamics of the activation of the BX-C locus? Is the transition gradual or immediate? Is it possible to ChIP some of the transcription factors thought to activate these genes*?

Unfortunately, the transgenes marking single parasegments do not turn on until at least 4 hours of development, well after repression via Pc-G has been established. These technical limitations make addressing this interesting developmental question in single parasegments quite challenging, and outside the scope of the current manuscript.

*What is the 3D conformation of the parasegments within the BX-C cluster? How do they compare to each other? Are active and repressive domains spatially partitioned*?

Members of our lab have performed 4C with three different baits in PS7 (*Abd-B* promoter, the *Ubx* promoter, and the border between PS7 and PS8 regulatory domains). We have not observed contacts changing in an obvious manner that correlate with domain activation and repression. Given the significant amount of sorted material required for the experiments and the absence of robust differences between mixed cells and PS7, we have chosen to save a more thorough investigation of these data for another manuscript.

*How is the expression and H3K27 distribution altered in mutants (null or hypomorphs) of Polycomb genes? Is there simply a decrease in H3K27me3, or is the H3K27me3 boundary shifted as well*?

We have developed the techniques to perform ChIP-seq from small numbers of precisely staged mutant embryos, and have begun to generate data that answer these questions. An analysis as thorough as this topic deserves is outside the scope of this work, and will be featured in a future manuscript.

Figure 3:

*How does the distribution of H2AK119Ub correlate with Polyhomeotic? Does it form blocks within transcriptionally repressed chromatin, as is the case for PRC2*?

We are also interested to the answer to this question. Unfortunately, the two commercially available antisera for mammalian H2AK199Ub (monoclonal E65C, and cell signaling 8240) failed to enrich BX-C chromatin under our experimental conditions. This may be because the amino acid residues flanking the ubiquitinated lysine residue are different in mammalian and Drosophila H2A (as suggested in Scheuermann et al., Fly 2012), and a fly-specific antibody is necessary for ChIP.

Reviewer #3:

*1) The authors should analyze the PRC2 binding profile (e.g. ChIP with antibody against E(z))in PS4, 5, 6 and 7, or at the minimum, just in PS7 as they did for Ph, CTCF etc.*.

**Experimental response:** ChIP-seq for Su(z)12 in mixed cells and PS7.

**Location in revised manuscript:**
Figure 3.

We found that PRC1 components bind to PREs regardless of their activation state, and the PRC2 component Su(z)12 looks similar.

*This would address the conceptually important point whether the binding of PRC2 at PREs is constitutive (i.e. PRC2 bound not only in the inactive but also in the active state; see also below) and that the control takes place at the level of deposition of the modification. Given that the authors may already have chromatin from PS7 nuclei or can generate it as they did for several of their experiments in the manuscript, I do not think that it is asking too much to perform such a ChIP experiment*.

*2) This point is directly connected to the first point. The authors must discuss their findings in the context of the findings from earlier studies (i.e. Papp and Müller, G&D 2006; Kwong et al, PLOS Genetics 2008; Soshnikova and Duboule, Science 2009)*.

Thanks for pointing out this oversight. In the revised manuscript, we have included additional references to [35], [22], and [45].